# Clinical Manifestation, Auxiliary Examination Features, and Prognosis of GFAP Autoimmunity: A Chinese Cohort Study

**DOI:** 10.3390/brainsci12121662

**Published:** 2022-12-03

**Authors:** Lei Liu, Boyan Fang, Zhixin Qiao, Xiaomeng Di, Qiuying Ma, Jingxiao Zhang, Jiawei Wang

**Affiliations:** 1Department of Neurology, Beijing Tongren Hospital, Capital Medical University, Beijing 100176, China; pathologyliu@163.com (L.L.); echotee@163.com (X.D.); maqiuyinglf@163.com (Q.M.); jingxiaobb@163.com (J.Z.); 2Neurological Rehabilitation Center, Beijing Rehabilitation Hospital, Capital Medical University, Beijing 100144, China; 3Medical Research Center, Beijing Tongren Hospital, Capital Medical University, Beijing 100730, China; zhixinqiao@live.com

**Keywords:** glial fibrillary acidic protein, autoimmunity, meningoencephalomyelitis, tremor, coexisting autoantibodies, diffuse large B-cell lymphoma, immunotherapy

## Abstract

Objective: This paper reports the clinical manifestation and auxiliary examination features of 15 Chinese patients with glial fibrillary acidic protein (GFAP) autoimmunity. Methods: From June 2016 to December 2019, patients suspected to have neurological autoimmune disease after having their serum and cerebrospinal fluid (CSF) tested for conventional neural antibodies were scanned for additional autoantibodies by immunohistochemistry. Samples that showed a characteristic immunoreactive pattern reminiscent of the GFAP of astrocytes were selected and confirmed by cell-based assay using cells-expressing human GFAPα. Results: A total of 15 patients (eight male and seven female) with a median age at onset of 53 years (range 28–72) were identified as GFAP-IgG-positive. Fourteen cases had GFAP-IgG detected in the CSF, while serum GFAP-IgG was detected in 11 cases. Eleven of the fifteen patients (73.3%) presented with an acute monophasic course, of which 10 (90.9%) had antecedent flu-like symptoms. The predominant phenotype was meningoencephalitis (46.7%), followed by meningoencephalomyelitis in 40% of the cases. The most common clinical features included long tract signs, brainstem symptoms, tremors, headaches, and psychiatric symptoms. Magnetic resonance imaging (MRI) revealed the enhancement of the meninges, the surface of the brainstem, the cerebellum, and the spinal cord as predominant. Inflammatory CSF showed mild lymphocyte-predominant pleocytosis with a median of 51/μL and elevated protein with a median of 87.5 mg/dL. Five patients had coexisting antibodies, including NMDAR-IgG in three patients and Yo and MOG-IgG in one patient each. One patient underwent a stereotactic brain biopsy, and the neuropathology diagnosis was diffuse large B-cell lymphoma. One patient had ovarian teratoma. Eleven of the fifteen (73.3%) patients received both intravenous immunoglobulin and steroids. Among them, three patients also received immunosuppressive agents later. During a two-year follow-up, 9 of the 15 (60%) patients achieved complete clinical remission. Conclusions: The clinical presentation of GFAP astrocytopathy is heterogeneous. It can be characterized by an acute monophasic course and a chronic relapsing course. Tremors are a prominent clinical manifestation in patients with an acute monophasic course with GFAP-IgG antibodies only. Most patients responded well to immunotherapy. In patients with GFAP autoimmunity, presenting with a chronic relapsing course, one should actively search for immunogenic factors and the culprit antibodies. In the case of primary central nervous system lymphoma, GFAP autoimmunity does not always equate to autoimmune GFAP astrocytopathy.

## 1. Introduction

First defined in 2016 [1], autoimmune glial fibrillary acidic protein (GFAP) astrocytopathy is a corticosteroid-responsive inflammatory central nervous system (CNS) disorder spectrum, predominantly affecting the meninges, brain, spinal cord, and optic nerves with manifestations of fever, headaches, tremors, ataxia, encephalopathy, myelitis, involuntary movement, and abnormal vision, mimicking infectious meningoencephalitis [2,3]. IgG antibodies binding to GFAP in the serum or the cerebrospinal fluid are specific biomarkers with diagnostic significance but without a pathogenic role. Neuropathological findings also indicated that lymphocytic infiltration substantially consisted of CD8 + T lymphocytes [4,5]. Autoimmune GFAP astrocytopathy may be accompanied by some autoantibodies, including anti-N-methyl-D-aspartate receptor (NMDAR) antibodies, anti-aquaporin4 (AQP4) antibodies, and anti-myelin oligodendrocyte glycoprotein (MOG) antibodies [6]. Other concomitant antibodies reported include anti-Purkinje cell cytoplasmic antigens, leucine-rich glioma-inactivated protein 1 (LGI1), contactin-associated protein 2 (CASPR2), glutamic acid decarboxylase 65 isoform (GAD65), type A receptors for gamma-aminobutyric acid (GABA_A_R), and GM3- and GM4-gangliosides [5,7,8]. As some cases are accompanied by tumors, it is suggested that GFAP autoimmunity may be the result of a paraneoplastic immune response initiated by a tumor that generates antibodies that cross-react with neuronal and glial cells [9]. However, unlike typical cytotoxic T cells that mediate paraneoplastic syndrome, autoimmune GFAP astrocytopathy patients generally have favorable corticosteroid responses [3]. Here, we retrospectively summarize the clinical and auxiliary examination characteristics and prognosis of 15 Chinese patients to provide additional findings on GFAP autoimmunity.

## 2. Patients and Methods

### 2.1. Patients

From June 2016 to December 2019, paired serum and cerebrospinal fluid (CSF) samples from 280 patients were gathered after tests for conventional neuronal or glial autoantibodies were further evaluated for GFAP-IgG in the medical research center of Beijing Tongren Hospital, Capital Medical University. Detailed demographic data, clinical manifestations, imaging, laboratory examinations, treatments, and the two-year follow-ups of patients with GFAP autoimmunity were retrospectively evaluated. All the patients underwent a comprehensive examination. Tests related to systemic autoimmune diseases (ESR, CRP, ANA, anti-DS DNA, anti-RNP, anti-SSA/SSB, anti-Scl-70, anti-Jo-1, anti-Sm, ANCA, and anti-TPO), suspicious infectious organisms, and tumor markers were thoroughly screened. The modified Rankin scale (mRS) score was applied to assess the treatment outcomes. This retrospective study was approved by the Ethics Committee of Beijing Tongren Hospital, Capital Medical University. All the patients provided written informed consent.

### 2.2. Measurement of Neuronal or Glial Autoantibodies

(1)An indirect immunofluorescence cell-based assay (CBA) using human embryonic kidney (HEK) 293 cells transfected with an appropriate expression of plasmids was used to detect conventional neuronal surface and glial antibodies targeting N-methyl-D-aspartate receptors (NMDARs), α-amino-3-hydroxy-5-methyl-4-isoxazolepropionic acid receptors 1/2 (AMPARs 1/2), gamma-aminobutyric acid type B receptors (GABA_B_Rs), leucine-rich glioma-inactivated protein 1 (LGI-1), contactin-associated protein 2 (CASPR2), aquaporin 4 (AQP4), and myelin oligodendrocyte glycoprotein (MOG) (FA112d-1005-1, FA1128-1005-50, FA1156-1005-50, Euroimmun AG, Lübeck, Germany);(2)A line immunoblot assay was used to detected antibodies targeting GAD 65, Titin (MGT30), Recoverin, PKCɣ, Zic4, Tr (DNER), SOX1, Ma2, Ma1, Amphiphysin, CV2 (CRMP5), Ri, Yo, and HuD associated with paraneoplastic neurological syndromes (PNS14-003, ravo Diagnostika GmbH, Freiburg, Germany);(3)All the serum and CSF specimens were further evaluated by indirect immunofluorescence tissue-based assay (TBA) using rat and monkey brain tissue (Euroimmun AG, Lübeck, Germany) for additional autoantibodies;(4)An indirect immunofluorescence cell-based assay (CBA) using human embryonic kidney (HEK) 293 cells transfected with GFAPα expression plasmids (MT231-16, Shaanxi MYBiotech Co. Ltd., Xian, China) was used to confirm whether the antibodies in selected samples present a characteristic immunoreactive pattern reminiscent of GFAP of astrocytes.

### 2.3. Statistical Analysis

Data are reported as the median and range or the number and percentage.

## 3. Results

### 3.1. Serum and CSF GFAP-IgG Status

Sixteen cases showed a GFAP-IgG staining pattern in the indirect immunofluorescence TBA. When confirmed with CBA, only one patient was negative and clinically diagnosed with multiple sclerosis. Among the 15 patients ultimately included, 14 cases had GFAP-IgG detected in their CSF, while 11 cases showed serum GFAP-IgG.

### 3.2. Clinical Characteristics of Patients with GFAP Autoimmunity

The demographics and clinical and paraclinical characteristics of the 15 patients are summarized in Table 1. The median age of the 15 patients was 58 years (range: 28–72), and eight (53.3%) were male. Ten (66.7%) patients were over 45 years old. Eleven of the fifteen patients (73.3%) presented with a monophasic course, of which nine had antecedent flu-like symptoms. Four patients (26.7%) had a recurrent course, of which two had prodromal infectious symptoms. One patient had a combination of ovarian teratoma. Primary central nervous system lymphoma (PCNS) was found in another patient. No systematic autoimmune diseases and no organ-specific antibodies were found in this group of patients. The clinical phenotypes included meningoencephalitis (seven patients, 46.7%), meningoencephalomyelitis (six patients, 40%), and two patients only with myelitis. The top five clinical findings included long tract signs, brainstem dysfunctions, tremors, headaches, and psychosis. One-third of the patients presented memory deficits, blurred vision, ataxia, and bladder dysfunctions. In a small minority, seizures (13.3%) and peripheral neuropathy (13.3%) were also encountered (Table 2). 

### 3.3. Diagnostic Radiology Findings

Neuroimaging findings of the 15 patients are summarized in Table 3. For the cranial MRI, although 12 out of the 15 patients (80%) showed hyperintensity lesions on the T2-weighted images (T2WI) and the fluid-attenuated inversion recovery (FLAIR) images, but eight showed nonspecific small-vessel ischemic changes. Only four patients showed multiple abnormal hyperintensity MRI lesions in the cerebral white matter, cerebral lobe, thalamus, basal ganglia, cerebellum, brainstem, and optic nerve. For the gadolinium-enhanced brain MRI, 7 out of 11 patients (64%) showed abnormal enhancement images. Besides the enhancement of the lesions on T2WI/FLAIR, multiple linear enhancements were also found in the meninges, cerebellum, brainstem surfaces, and cerebral lobes. Two patients showed classic linear perivascular radial-enhancement patterns in the cerebral white matter. Four out of nine patients underwent spine MRI, showing intramedullary hyperintensity lesions on T2WI. Five out of the nine patients who underwent enhanced spinal MRI showed abnormal enhancement images. One patient showed an enhancement of the cervical and thoracic spinal cord surface, and 5 patients showed intramedullary enhancements at the cervical and/or thoracic spinal cord. Four patients underwent ^18^F-fluorodexyglucose positron emission tomography (^18^F-FDG PET) to screen for tumors. Two patients had low uptake in the frontal, parietal, temporal, or posterior lobes. One patient had low uptake in the left putamen. One patient showed high uptake in the frontal lobe, possibly affected by nearby meningitis. No concomitant tumor was found.

### 3.4. Laboratory Findings

Routine CSF analysis was performed for all 15 patients. Most patients tested had inflammatory CSF. The white blood cell count ranged from 5 to 300/μL, with a median of 51/μL. The protein level ranged from 40 to 177 mg/dL, with a median of 88 mg/dL. Six patients were tested for CSF-specific oligoclonal bands, of which five were positive.

### 3.5. Coexisting Autoantibodies and Oncological Associations

Five of the fifteen (30%) patients had coexisting antibodies detected in the serum or CSF, including CSF NMDAR-IgG in three patients (60%), serum Yo in one patient (20%), and serum MOG-IgG in one patient (20%). None of the fifteen patients had anti-aquaporin 4 (AQP4) antibodies. Case 6 underwent a stereotactic brain biopsy, and the neuropathology diagnosis was diffuse large B-cell lymphoma (DLBCL). In 10 patients’ CSF, only GFAP IgG was detected.

### 3.6. Treatment and Prognosis

For the treatment and prognosis of the 15 patients summarized in Table 1, 11 of the 15 (73.3%) patients received both intravenous immunoglobulin (IVIG) and steroids. Two patients were administered oral steroids only. One patient was treated with acyclovir only. Case 6 was not given any immunotherapy due to her primary central nervous system lymphoma. Only two patients received maintenance immunosuppressive agents (one with co-existing NMDAR antibody received azathioprine and one with co-existing MOG antibody received mycophenolate mofetil). The other patients had no subsequent immunosuppressive treatment after their acute-phase treatment. During the two-year follow-up, 9 of the 15 (60%) patients achieved complete clinical remission. Eight out of ten patients completely recovered when monitored at the two-year follow up, with no further subsequent immunosuppressive treatment. Four patients had an mRS score of 1. Among these four patients, one had NMDAR antibody coexisting with GFAP antibody, and one had MOG antibody coexisting with GFAP antibody. The other two patients were only administered with oral steroids and acyclovir. Case 11 had an mRS score of 4, which was also combined with positive NMDAR antibody. Case 6 was lost in the follow-up.

## 4. Index Cases

### 4.1. Case 2

A 62-year-old man was presented to the hospital for 11 days with progressive cognitive decline and reduced spontaneous speech and lethargy. He had a past medical history of hypertension and pemphigoid. On presentation, the patient displayed a mild fever, a stupor, and an inability to communicate and execute commands. Muscle strength was normal with echomotism, and the bilateral palmomental reflexes were positive. There was no nuchal rigidity. Cranial MRI showed T2 and fluid-attenuated inversion recovery (FLAIR) hyperintense signal changes in the bilateral corona radiata without gadolinium enhancement. Initial CSF analysis revealed lymphocytic pleocytosis and elevated protein. His CSF NMDAR-IgG was positive (titer 1:320), but his serum was negative. TBA showed GFAP staining patterns in his CSF, which were subsequently proved by CBA (titer 1:40)

The patient developed a generalized convulsion after admission. He was treated with IVIG, IVMP, valproate, and olanzapine, according to anti-NMDAR encephalitis. The seizures were controlled, and his cognitive function gradually improved. His MMSE (mini-mental state examination) returned to 29/30 within two months. 

### 4.2. Case 3

A 58-year-old woman was admitted to the hospital with fever, body aches, coughing (for 50 days), and limb tremors (for 30 days). Her maximum temperature was 38.5 °C. Her past medical history was unremarkable. On physical examination, she was awake, alert, and oriented to person, time, location, and situation. Her cranial nerves, sensory examination, and deep tendon reflexes were normal. The motor examination was notable for an increased tone on all the extremities with intact strength. Static tremor and postural tremor were present in the bilateral upper limbs. The cerebellum examination was normal. The CSF showed normal glucose, slightly elevated protein, and mild pleocytosis with a lymphocytic predominance. The cranial MRI revealed leptomeningeal enhancement in both the cerebellar (Figure 1A) and cerebral (Figure 1B) hemispheres on admission. The conventional autoimmune neurology panel testing for antibodies was negative. However, the TBA of the serum and CSF showed GFAP staining patterns in both monkey (Figure 1C) and rat (Figure 1D) brain tissue, which was later proved by specific CBA. The CSF cytology showed increased activated lymphocytes (Figure 1E). IVIG (0.4 g/kg body weight/day for 5 days) and IVMP (1000 mg/day for 3 days, then 500 mg/day for 3 days) were given followed by oral steroids. The CSF, before discharge, showed normal WBC counts and cytology (Figure 1F). The tapering of oral steroids resulted in the complete resolution of the symptoms after 6 months.

### 4.3. Case 5

A 59-year-old woman presented with fever for 43 days, hand shaking and defecation difficulties for one month, urinary retention for 10 days, and bilateral lower limb weakness for 1 week. Her maximum temperature was 38.5 °C. Her previous medical history was unremarkable. Upon physical examination, the patient was awake and alert but with slight dysarthria. She presented small horizontal nonpersistent nystagmus and tongue tremors. Increased muscle tone and tremors were noticed in the upper extremities. Her bilateral leg weakness (4/5 muscle strength) was accompanied by areflexia and bilateral Babinski signs. The cranial MRI was unremarkable except for FLAIR hyperintense signal changes in the bilateral corona radiata without enhancement. Although the cervical MRI revealed no abnormalities in the T2 sequence (Figure 2A), linear radial perivascular enhancement from the ventral to the dorsal pons was noticed, with salt-and-pepper-like enhancement in the cervical cord parenchyma (Figure 2B). Symmetric enhancement was observed bilaterally on the funiculus lateralis in both the upper thoracic segment (Figure 2C) and the lower thoracic segment (Figure 2D). Leptomeningeal enhancement was observed in the sagittal MRI of the lumbar spine (Figure 2E). She was treated with empiric acyclovir, antibiotics, and IVIG, but did not see clinical improvements (except in temperature). Her serum Yo antibodies were positive. The TBA and CBA revealed GFAP-IgG in both the serum and CSF. IVMP was given, and her tremors diminished completely within three days, and the muscle strength of her lower limbs improved. Her urinary retention was relieved gradually. The T2 (Figure 2G) and postcontrast T1 (Figure 2H) sequences of the cervical MRI were normal after steroids. The PET did not reveal occult tumors but rather increased metabolic activities at the cervical medulla and T12-L1 levels at the anterior spinal cord (Figure 2F). After tapering the oral steroids, she was symptom-free within a year. During the two-year follow-up, her serum Yo antibodies and GFAP-IgG in serum and CSF remained positive.

### 4.4. Case 6

A 35-year-old woman was admitted with shadow in her vision for 20 months and drowsiness and walking instability for 5 days. Twenty months previously, painless nasal defects happened in her right eye. Fluorescein fundus angiography suggested bilateral retina and choroid lesions. The cranial MRI with contrast showed no abnormalities. After two months of empiric treatment with vitamin B12, her symptoms improved spontaneously. Seventeen months previously, she complained of cobweb-like vision in her left eye. Pathology after vitrectomy revealed a small number of dysplasia cells. Clonal rearrangements of the IGH, IGK, TCRG, and TCRB genes were detected.

She refused chemotherapies for occult lymphoma. Five days previously, she was found with apathy and walking instability. On arrival, the patient was impassive and showed weakness and poor understanding. Her cranial nerves were intact. The motor examination was notable for the decreased muscle tone on all her extremities, but with normal strength and tendon reflex. Babinski signs were present bilaterally. Severe ataxia, dysmetria, and dysdiadochokinesia were also present. The cranial MRI showed mild leptomeningeal enhancement in the cerebellar hemisphere, multiple hyperintense T2, and enhanced lesions in the corpora quadrigemina, left dorsal pons, right thalamus, bilateral internal capsule, and left globus pallidus (Figure 3A–F). The CSF, TBA, and CBA proved the existence of GFAP-IgG (Figure 3G–I). The CSF flow cytometry showed abnormal CD4- and CD8-positive T cells but without abnormal CD19-positive B cells. A stereotactic brain biopsy of the left basal ganglia revealed diffuse large B-cell lymphoma without astrocytopathy (Figure 3J–N). After a clinicopathological diagnosis of primary central nervous system lymphoma was made, the patient was transferred to the hematology department for chemotherapy.

Two other cases (Cases 7 and 8) in this cohort have already been reported [10,11].

## 5. Discussion

In 2016, Fang and Lennon first described GFAP-IgG, an antibody associated with autoimmune meningoencephalomyelitis, and unified the spectrum of diseases under the name of autoimmune GFAP astrocytopathy [1]. GFAP is the main intermediate filament protein almost exclusively expressed in astrocytes [12] and is involved in the maintenance of cytoskeletal structures and functions and in various pathological processes, such as trauma, stroke, and neurodegenerative diseases. GFAP mutations result in Rosenthal fibers in Alexander disease [13]. To date, there are seven isoforms of GFAP proteins, including α, β, γ, δ/ε, and κ, which are already known to be expressed in the CNS and participate in various physiological and pathological processes.

Previous studies have revealed that autoimmune GFAP astrocytopathy usually happens in individuals over 40 years old [2,8]. Most cases have an acute or subacute onset, with heterogeneous clinical manifestations, including fever, headache, encephalopathy, movement disorders (tremor, myoclonus, and ataxia), autonomic dysfunction (predominantly urinary retention), myelitis, and vision loss. Furthermore, area postrema syndrome (APS) has been previously described [14]. The majority of our patients with a monophasic course had a tendency towards prodromal flu-like symptoms before onset. Meanwhile, Flanagan’s investigation demonstrated that 29% of patients (11/38) had prodromal flu-like symptoms before onset [2]. Two patients with an acute onset accompanied by NMDAR-IgG antibodies had clinical, electroencephalographic, and CSF features consistent with autoimmune encephalitis, but apathy was prominent therein. Another patient with a chronic recurrent course had manifestations of encephalitis in the early stage with significant psychiatric symptoms and multiple intracranial lesions with gadolinium enhancement and was NMDAR-IgG-positive but GFAP-IgG-negative in the CSF. With his disease fluctuation, CSF NMDAR-IgG turned negative, and his GFAP-IgG levels gradually elevated from moderately positive to prominently positive. His mental status shifted to hypersomnia and apathy. Another symptom often described but less discussed and prevalent in our cohort is that of tremors. Tremors are the predominant symptom in eight out of ten patients who only have GFAP antibodies. Among the signs and symptoms listed in our table, although tremors are behind the long-tract signs and brainstem signs, they should be treated as a key diagnostic clue because of their easy detection. The underlying pathogenesis may be relevant to the disturbance of the homeostatic control of dopamine in astrocytes [15]. However, other types of movement disorders, such as myoclonus and dyskinesia reported in a Japanese group [16,17], were not found in our cohort.

Previous studies revealed that 88% of autoimmune GFAP astrocytopathy patients had elevated CSF white blood cell counts, 83% had elevated protein levels, and 50% had positive specific oligoclonal bands [2]. Similarly, a recent French cohort study showed that 98% of patients had leukocytosis in the CSF, and 93% had elevated protein [18]. The CSF studies of our cohort confirmed these findings. Coexisting neuronal autoantibodies could be detected in the serum or CSF of up to 30–40% of autoimmune GFAP astrocytopathy patients, according to a previous study. The most common coexisting antibodies are NMDA-R-IgG and aquaporin-4 (AQP4)-IgG [2,3]. But the Japanese study [17] and French study [18] had different observations, with a smaller proportion of these two antibodies. Our study found that 30% of the patients had coexisting antibodies that were detected in the serum or CSF; the most common was NMDAR-IgG, which is consistent with previous studies. But the proportion of concurrent ovarian teratomas was found to be significantly lower than that noted in previous research.

Lesions of autoimmune GFAP astrocytopathy may involve the basal ganglia, hypothalamus, brainstem, cerebellum, spinal cord, and subcortical white matter [16]. Although radial perivascular emphasis can be seen in other diseases [19], approximately 56% of the patients showed T2 hyperintensities via cranial MRI, and approximately 56% showed linear enhancements around the white matter vessels perpendicular to the ventricles, while only a few patients showed no abnormalities [3]. Spinal cord lesions are commonly longitudinally extensive and predominantly involve the central gray matter of the thoracic spinal cord [8]. In our cohort, patients with linear perivascular radial gadolinium enhancement in the white matter perpendicular to the ventricle were extremely rare, while cerebellar leptomeningeal enhancement and longitudinally extensive lesions of the spinal cord with spinal meninges enhancement were noteworthy. In the T2WI and FLAIR sequences, although there were many abnormal hypersensitive signals, most of them were nonspecific. The enhancement images could provide more positive information, especially the enhancement of the meninges, the surface of the brainstem, the cerebellum, and the spinal cord. Electrophysiological examination revealed radiculopathy in two patients.

The majority of patients are steroid-responsive, but there are also a few individuals who do not improve after receiving corticosteroids and then face variable extents of disability, relapse, or even death [3,20]. A lack of response to steroids reminds us of coexisting neuronal autoantibodies or malignancy. Second-line immunosuppressive agents, such as mycophenolate mofetil, azathioprine, rituximab, and cyclophosphamide, could be considered for refractory or relapse cases. Most patients in our study responded well to corticosteroids and barely suffered relapses during the follow-up, which parallels previous research [3]. One patient with MOGAD was treated with mycophenolate mofetil to prevent the recurrence of optic neuritis and myelitis [11]. One patient with anti-NMDAR encephalitis was treated with azathioprine to prevent relapse.

To date, the etiology of GFAP astrocytopathy has remained obscure. However, similar to autoimmune encephalitis, there is evidence suggesting that it may be associated with tumors and infections. In our study, one case was accompanied by ovarian teratoma [11]. PCNSL was found in Case 6. Another case of GFAP-IgG that was coexistent with PCNSL has been reported this year, which raised two important issues [21]. Firstly, GFAP autoimmunity may not always be a remote effect of the neoplasm but could also happen intrathecally. Secondly, GFAP-positive fibrillary astrocytes mingling with tumor cells without clasmatodendrosis suggested that GFAP autoimmunity does not equate to autoimmune GFAP astrocytopathy [18]. This reminds us that PCNSL should be cautiously excluded before making the diagnosis of GFAP astrocytopathy.

Initial studies of GFAP-IgG focused on the α and δ/ε isoforms, and later, studies then shifted emphasis to the detection of GFAP-α antibodies in the cerebrospinal fluid. Although GFAP-IgG in the CSF was recognized as the fifth most common autoimmune encephalitis biomarker [22], the role of this antibody in the mechanism of this disease is still obscure. Antibody-associated encephalitis caused by the analogous paraneoplastic intracellular antigens, such as Hu, Yo, Ri, Tr, and CV2/CRMP5, which were first identified, was found to have primary pathogenesis mediated by cytotoxic T cells. Most scholars believe that GFAP-IgG, an intracellular antigen, is probably not directly pathogenic but rather a biomarker of the cytotoxic T-cell-mediated autoimmune response [23].

## 6. Conclusions

Autoimmune GFAP astrocytopathy can be characterized by an acute monophasic course and a chronic relapsing course. Tremors are a prominent clinical manifestation in this group of patients, especially in patients with an acute monophasic course with GFAP-IgG antibodies only. GFAP-IgG antibodies can coexist with anti-NMDAR-IgG, MOG-IgG, and Yo antibodies, but the proportion of concurrent ovarian teratomas was found to be significantly lower than that noted in previous research. Our patients with an acute monophasic course and with antecedent infection responded well to immunotherapy. In patients with GFAP-IgG-associated disease presenting with a chronic relapsing course, we should focus on actively searching for immunogenic factors or the responsible antibodies. In particular, we should pay extra attention to concomitant primary CNS lymphoma. Thus far, the whole picture of GFAP autoimmunity is still not clear. The clinical manifestations of GFAP autoimmunity are heterogeneous among different cohorts, possibly due to the small samples or the human species, and require further study.

## Figures and Tables

**Figure 1 brainsci-12-01662-f001:**
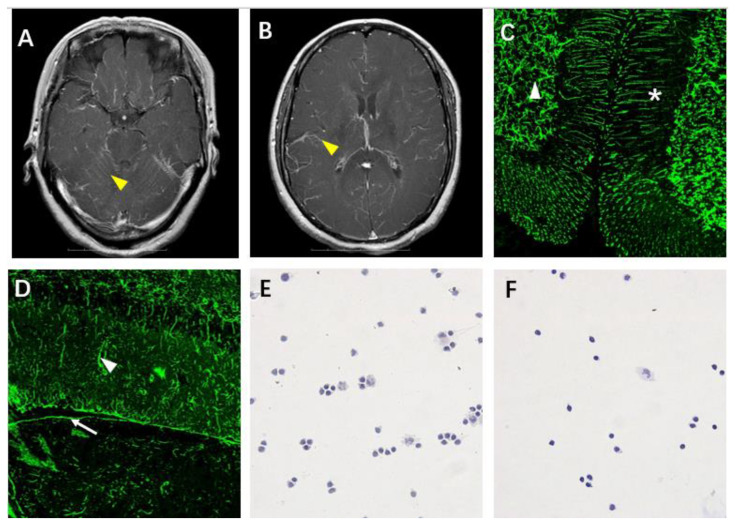
Neuroimaging, serological, and CSF cytological findings of Case 3. Cranial MRI on admission revealed leptomeningeal enhancement in both cerebellar (**A**) and cerebral (**B**) hemispheres (yellow arrowhead). TBA of CSF showed characteristic Bergmann radial pattern in the molecular layer (asterisk) and astrocytic processes in the white matter (white arrowhead) of monkey cerebellum (**C**). IgG also bound pia (white arrow) and astrocytic processes (white arrowhead) in the molecular layer of rat hippocampus (**D**). CSF cytology showed increasing activated lymphocytes during acute phase (**E**). CSF cytology before discharge with steroids became normal (**F**).

**Figure 2 brainsci-12-01662-f002:**
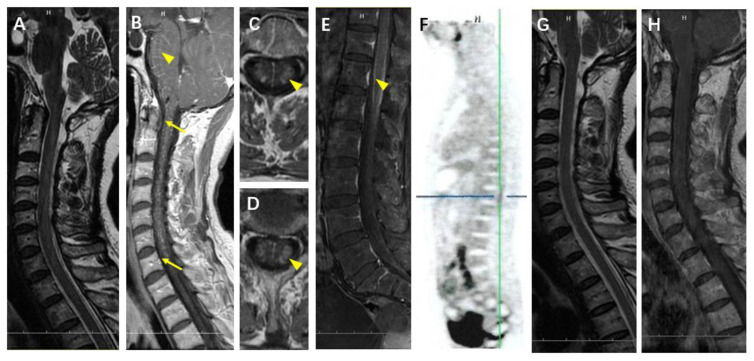
Radiological features of Case 5. On admission, cervical MRI revealed no abnormalities in the T2 sequence (**A**); however, linear radial perivascular enhancement from the ventral to the dorsal pons was noticed (arrowhead), with salt-and-pepper-like enhancement in the cervical cord parenchyma (arrow) (**B**). Symmetric enhancement (arrowhead) was observed bilaterally on the funiculus lateralis in both the upper thoracic segment (**C**) and the lower thoracic segment (**D**). Leptomeningeal enhancement (arrow head) was present in the sagittal MRI of the lumbar spine (**E**). PET revealed increased metabolic activity at the T12-L1 levels of the anterior spinal cord (**F**, crossing lines). The T2 (**G**) and postcontrast T1 (**H**) sequences of the cervical MRI were normal after steroids.

**Figure 3 brainsci-12-01662-f003:**
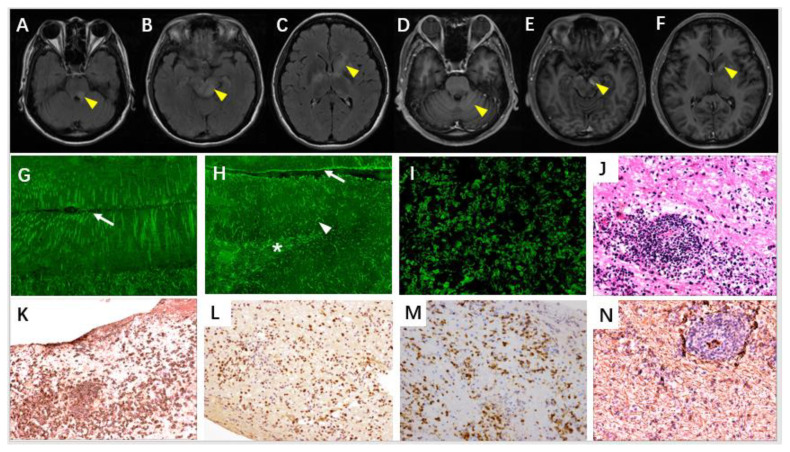
Neuroimaging, serological, and neuropathological alterations in Case 6. Cranial MRI showed multiple hyperintense T2-FLAIR (**A**–**C**) and T1 enhanced (**D**–**F**) lesions in the corpora quadrigemina (**B**), left dorsal pons (**A**), left cerebral peduncle (**E**) and left caudate (**C**,**F**). There was mild leptomeningeal enhancement in the cerebellar hemisphere (**D**) (arrow head). TBA on rat cerebellum cortex with patient’s CSF showed Bergmann glial short radial staining (arrow) in two parallel molecular layers separated by a sulcus (**G**). TBA on rat hippocampus with CSF demonstrated pia (arrow), cortical molecular layer (arrow head), and white matter astrocytic processes (asterisk) positive (**H**). GFAP-IgG in CSF was further identified through a commercial CBA of HEK293 cells transfected with GFAPα expression plasmids (**I**). Biopsy of her left basal ganglia showed tumor cells around the vessel wall with lymphocytic cuffing (**J**: hematoxylin–eosin). Tumor cells, labeled with CD20 (**K**), Bcl-6 (**L**), and Mum-1 (**M**) in immunohistochemistry, enabled the diagnosis of diffuse large B-cell lymphoma. Notably, the reactive fibrillary astrocytes mingling with tumor cells were GFAP-positive with intact processes (**N**) (**G**–**N** ×200).

**Table 1 brainsci-12-01662-t001:** Demographic, clinical, radiological, and immunological characteristics of 15 patients with GFAP autoimmunity.

No.	Sex/Age	Monophasic	Prodromal Symptoms	Clinical Manifestations	CSF Findings	GFAP-IgG	Other Abs	Tumors	MRI Enhancement	Immunotherapies	Two-Year Follow Up (mRS)
1	M/62	Yes	Flu-like	Headache, diplopia, ptosis	WBC 5, Pro 66 mg/dL, SOB (+)	CSF+/Serum+	None	No	No	IVIG + Oral MP	Complete recovery
2	M/62	Yes	Flu-like	Cognitive decline, epilepsy	WBC 42, Pro 115 mg/dL	CSF+/Serum−	CSF: NMDAR	No	No	IVIG + IVMP	Complete recovery
3	F/58	Yes	Flu-like	Tremor, hypertonia	WBC 140, Pro 67 mg/dL	CSF+/Serum+	None	No	Yes	ACV + IVIG + IVMP	Complete recovery
4	M/59	Yes	Flu-like	Urinary retention, tremor, facial twitching	WBC 45, Pro 88 mg/dL, SOB (+)	CSF+/Serum−	None	No	NA	IVIG + Oral MP	Complete recovery
5	F/59	Yes	Flu-like	Urinary retention, tremor, hypertonia	WBC116, Pro 120 mg/dL, SOB(+)	CSF+/Serum+	Serum: Yo *	PET (−)	Yes	ACV + IVIG + IVMP	Complete recovery
6	F/35	Multiphasic	None	Ataxia, blurry vision	WBC 6, Pro 46.4 mg/dL, SOB(+)	CSF+/Serum−	None	PCNSL, PET (−)	Yes	No	Lost
7	F/43	Multiphasic	None	Myelitis, epilepsy, ON	NA, SOB(-)	CSF-/Serum+	Serum: MOG	OT	Yes	IVIG + IVMP + MMF	mRS 1
8	F/28	Yes	Flu-like	Tremor, ataxia	WBC 300, Pro 177 mg/dL	CSF+/Serum+	No	No	Yes	IVIG + Oral MP	Complete recovery
9	M/63	Yes	Flu-like	Tremor, radicular pain, abnormal behavior, memory loss, ataxia	WBC 10, Pro 68 mg/dL	CSF+/Serum+	No	PET (−)	No	Oral MP	Complete recovery
10	M/48	Multiphasic	Flu-like	Diplopia, altered mental status, slow response, aphasia, hypersomnia	WBC 21, Pro 40 mg/dL	CSF+/Serum+	CSF: NMDAR	No	Yes	IVIG + IVMP + AZA	mRS 1
11	F/72	Yes	Flu-like	Cognitive decline, apathy, urinary retention	WBC 51, Pro 103 mg/dL, SOB(+)	CSF+/Serum+	CSF: NMDAR	No	Yes	IVIG + IVMP	mRS 4
12	M/52	Yes	None	Dizziness, tremor, myelitis	WBC 25, Pro 77 mg/dL	CSF+/Serum−	No	No	Yes	IVIG + IVMP	Complete recovery
13	M//39	Multiphasic	Flu-like	Urinary retention, blurry vision, peripheral neuropathy	WBC 233, Pro 99 mg/dL	CSF+/Serum+	No	No	Yes	Oral MP	mRS 1
14	M/59	Yes	None	Tremor	WBC 87, Pro 101 mg/dL	CSF+/Serum+	No	PET (−)	NA	ACV	mRS 1
15	F/41	Yes	Flu-like	Tremor, urine retention, constipation	WBC 100, Pro 130 mg/dL	CSF+/Serum+	No	No	Yes	ACV + IVIG + IVMP	Complete recovery

Abbreviations: Abs, antibodies; ACV, acyclovir; AZA, azathioprine; CSF, cerebrospinal fluid; F, female; GFAP, glial fibrillary acidic protein; IgG, immunoglobulin G; IVIG, intravenous immunoglobulin; IVMP, intravenous methylprednisolone; M, male; MMF, mycophenolate mofetil; MOG, myelin oligodendrocyte glycoprotein; MP, methylprednisolone; MRI, magnetic resonance imaging; mRS, modified Rankin Scale; NA, not available; NMDAR, N-methyl-D-aspartate-receptor; No., number; ON, optic neuritis; OT, ovarian teratoma; PCNSL, primary central nervous system lymphoma; PET, positron emission tomography; Pro, protein; SOB, specific oligoband; WBC, white blood cell. * This case with Yo antibody was positive in the serum immunoblot but was subsequently confirmed negative by TBA. No tumor was found in the follow-up.

**Table 2 brainsci-12-01662-t002:** Clinical phenotypes and detailed clinical findings of 15 patients with GFAP autoimmunity.

No.	Clinical Phenotype	Long Tract Signs	Brainstem Signs	Tremor	Headache	Psychiatric Symptoms	Memory Deficits	Blurred Vision	Ataxia	Bladder Dysfunction	Seizures	Radiculoneuropathy
1	ME	+	+		+			+				
2	ME					+	+				+	
3	ME			+	+					+		
4	MEM		+	+	+				+	+		+
5	MEM	+	+	+	+					+		
6	ME	+	+			+		+	+			
7	MEM	+						+		+	+	
8	MEM	+		+	+	+	+	+	+			
9	MEM	+	+	+	+	+	+		+			+
10	ME		+		+	+	+					
11	ME	+				+	+					
12	M		+	+					+			
13	MEM	+	+					+				
14	ME	+	+	+								
15	M	+		+						+		
Total	2M:7ME:6MEM	10	9	8	7	6	5	5	5	5	2	2

Abbreviations: M, myelitis; ME, meningoencephalitis; MEM, meningoencephalomyelitis.

**Table 3 brainsci-12-01662-t003:** Neuroimaging findings of the 15 patients with GFAP autoimmunity.

Examination	Patients, Number (%)
Cranial MRI	15
Abnormal hyperintensity lesions on T2WI/FLAIR images	12 (80)
Cerebral white matter	3 (25)
Cerebral lobe	1 (8.3)
Thalamus	1 (8.3)
Basal ganglia	2 (17)
Cerebellum	2 (17)
Brainstem	2 (17)
Optic nerve	1 (8.3)
Nonspecific small vessel ischemic changes	8 (67)
Gadolinium-enhanced brain MRI	11
Abnormal enhancement images	7 (64)
Linear radial enhancement pattern of cerebral white matter	2 (18)
Enhancement of meninges	4 (36)
Cerebellum	4 (36)
Brainstem	3 (27)
Cerebral lobe	2 (18)
Thalamus	2 (18)
Basal ganglia	2 (18)
Spine MRI	9
Intramedullary hyperintensity lesions on T2WI	4 (44)
Gadolinium-enhanced spine MRI	5
Abnormal enhancement images	5 (100)
Enhancement of spinal cord surface	1 (20)
Intramedullary enhancement	5 (100)
^18^F-FDG PET	4
Low uptake	3 (75)
Brain lobe (Frontal, Parietal, Temporal, Posterior lobe)	2 (50)
Putamen	1 (25)
High uptake	2 (50)
Frontal lobe	1 (25)

GFAP: glial fibrillary acidic protein; MRI: magnetic resonance imaging; T2WI:T2-weighted image; FLAIR: fluid-attenuated inversion recovery; ^18^F-FDG PET:^18^F-fluorodexyglucose positron emission tomography.

## Data Availability

All the data reported within the article are anonymized and available upon request from the qualified investigators.

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
