# Peer review of "Clinical Manifestation, Auxiliary Examination Features, and Prognosis of GFAP Autoimmunity: A Chinese Cohort Study"

_brainsci, 2022, doi:10.3390/brainsci12121662_

Round 1
Reviewer 1 Report
Dear colleagues,
thank you for presenting your findings. Although it was a pleasure reading your draft, I suggest that you shorten abstract and case reports. In figure 2, image H should be shifted behind image E.
Kind regards and best wishes!
Author Response
Response to Reviewer 1 Comments:
Point 1. thank you for presenting your findings. Although it was a pleasure reading your draft, I suggest that you shorten abstract and case reports.
Response 1:
Thank you for your kind suggestions.
We have tried hard to shorten the abstract and case reports to a certain extent while preserving completeness. Thank you so much!
Point 2. In figure 2, image H should be shifted behind image E.
Response 2:
Thank you for your kind suggestions. We put the PET-CT (the previous image H) behind image E as you kindly advised.

Reviewer 2 Report
The goal of the MS by Liu et al. is to report the cases of GFAP autoimmunity in 15 Chinese patients. It is a retrospective cohort study comprising 8 male and 7 female patients that visited the Beijing Tongren Hospital (Capital Medical University) from June 2016 until December 2019.
The Introduction section contains an adequate amount of information on the state-of-the-art. Even though the Materials and Methods section is well conceptualized, there is very little to no data on the examined patients. Even though GFAP autoimmunity is a novel entity, there are already some notable advancements with respect to the concomitant occurrence with preexisting conditions. For example, a paper by Gravier-Dumonceau (DOI: 10.1212/WNL.0000000000013087) reports the occurrence of one or more coexisting autoimmune disorders in patients which also presented with GFAP autoimmunity. As such, one of the major flaws of this MS is the lack of patient data. Even though the authors did include some important clinical characteristics of patients with GFAP autoimmunity within the Results section (3.2), there is no information on existing comorbidities. Therefore, to better assess the instances of the occurrence of GFAP autoimmunity, the authors need to add other information as it pertains to the examined patients. Most notably, an analysis of the existing comorbidities and a comment on those with respect to other case studies on GFAP autoimmunity needs to be added.
The Results section contains a good graphical depiction of some characteristics of the observed patients in Table 1, as well as the associated clinical phenotypes in Table 2. This represents a good addition to the MS and will be valuable to the readers. Even though the authors did include a three-sentence comment on diagnostic radiology findings (3.3) within this section, this is insufficient. The authors do attempt to list the regions abnormal enhancement was noted in gadolinium-enhanced MRI, but the phrasing of the second sentence in this paragraph is very poor. Additionally, was any spine MRI performed? What were the diagnostic readings like? Since the authors do appear to have extensive information about the diagnostic and radiology findings, as seen in section Index cases (4), I suggest that the authors compile all of these readings (both from the patients mentioned within the MS and those not included here) and incorporate them into the section 3.3, be it in the form of a table or within the text itself. (For reference see Table 5 in DOI: 10.1016/j.jneuroim.2019.04.004).
Next, even though the authors list the treatment and prognosis within the Results section (3.6), there is little information on the follow-up. Was any maintenance therapy dispensed? Including oral corticosteroids, rituximab, cyclophosphamide, mycophenolate mofetil or azathioprine? Additionally, what was the procedure like with patients that did not achieve remission and needed additional treatment?
Finally, the manuscript should undergo major editing of the English language and style since it significantly impacts the legibility of the MS. This includes a multitude of sentences with no verbs and unclear wording. As such, and due to many omissions within the MS, I do not recommend the publication of this MS in Brain Sciences in its current form, but rather suggest for it to be reconsidered following major revisions.
Reviewer 3 Report
The authors report a single institution case series of autoimmune GFAP astrocytopathy. Demographics, clinical presentation, laboratory and imaging data, course and response to therapy are reported. Many of the reported features are in agreement with prior literature. A single case of CNS lymphoma with CSF anti-GFAP IgG is reported, which may be novel, but possibly an incidental finding. Thus overall, this report represents incremental advance in knowledge.
Round 2
Reviewer 2 Report
The authors have meticulously addressed of my concerns and the MS has undergone extensive editing of the English language and style. The authors have included additional details on patients’ comorbidities and conducted medical examinations as well as discussed the findings on coexisting antibodies with respect to other case studies. Additionally, the authors have also summarized the diagnostic radiology findings in greater detail within the text of the Results section as well as the newly added Table 3. Both of these are a great addition to the manuscript. What is also greatly appreciated is the additional elaboration on maintenance therapy administered to each patient during the follow-up. As such, and after performing the suggested revisions, Liu et al. have significantly improved the manuscript which warrants its publication in Brain Sciences.